# Detecting deception via eyeblink frequency modulation

Brandon S. Perelman

Michigan Technological University, Department of Cognitive and Learning Sciences, Houghton, MI, USA

## ABSTRACT

To assess the efficacy of using eyeblink frequency modulation to detect deception about a third party, 32 participants were sent on a mission to deliver a package to an interviewer. 17 of the participants lied to the interviewer about the details of their mock mission and 15 responded truthfully. During the interview, eyeblink frequency data were collected via electromyography and recorded video. Liars displayed eyeblink frequency suppression while lying, while truth tellers exhibited an increase in eyeblink frequency during the mission relevant questioning period. The compensatory flurry of eyeblinks following deception observed in previous studies was absent in the present study. A discriminant function using eyeblink suppression to predict lying correctly classified 81.3% of cases, with a sensitivity of 88.2% and a specificity of 73.3%. This technique, yielding a reasonable sensitivity, shows promise for future testing as, unlike polygraph, it is compatible with distance technology.

## INTRODUCTION

Modern deception detection methods testing physiological indices of deception use techniques such as galvanic skin response (GSR; e.g., *Carmel et al., 2003*), fMRI (e.g., *Bhatt et al., 2009*; *Langleben et al., 2005*; *Shah et al., 2001*), and EEG detected P300 event related potentials (*Abootalebi, Moradi & Khalilzadeh, 2006*; *Ambach et al., 2010*; *Meijer et al., 2007*). In the academic literature, responses are sometimes elicited (e.g., *Langleben, 2008*) using some variation of the Guilty Knowledge Test (GKT; *Lykken, 1957*; *Lykken, 1959*). Though these physiological techniques may provide strong sensitivity and specificity in laboratory settings (*MacLaren, 2001*), they require proprietary equipment, proximity to the suspect, and suspect awareness of the analysis. Furthermore, classification accuracy of GKT based tests obtained in laboratory settings from mock crime scenarios may not generalize to naturalistic settings (*Carmel et al., 2003*).

The present study explores a known behavioral indicator of deception, cognitive demand modulated blink frequency (BF). BF modulation is an attractive behavioral indicator of deception because BF data may be collected using hidden cameras or distance technology (i.e., web cams) and analyzed surreptitiously either in real time or post hoc from recorded video. Furthermore, BF data collection does not require any special equipment or questioning schemes. BF modulation has been experimentally validated using both GKT based questioning methods reliant upon recognition

Corresponding author
Brandon S. Perelman,
BPerelman@gm.slc.edu

(*Leal & Vrij, 2010*) and a conversationally natural free recall method (*Leal & Vrij, 2008*). The present study builds upon previous research using a mission based scenario in which the deception pertains to a third party, under conditions that more closely resemble an interaction using distance technology. In particular, the present study accounts for variance attributable to cognitive demand resulting from participants monitoring the interviewer's body language. As the literature regarding deception about a third party is somewhat sparse, the study expands prior work on third party deception to include validation using BF based measures of cognitive demand.

## Cognitive demand modulated BF and deception detection

The cognitive demand hypothesis follows that deception is more demanding than truth telling because, in addition to recall and speech production required during truth telling, deception requires suppressing deceptive cues in body language (*DePaulo, 1988*; *Ekman, 1989*), fabricating an alternate story, and carefully monitoring that story to ensure that it does not contradict interviewer knowledge of the event (*Leal & Vrij, 2008*). Cognitive demand during deception is detectable in experimental settings using measures such as response time (*Gronau, Ben-Shakar & Cohen, 2005*; *Seymour, Kerlin & Kurtz, 2003*) and startle response modulation (*Cacioppo, 2006*; *Verschuere et al., 2007*). Cognitive demand during deception is also observed in real high stakes police interviews (*Mann, Vrij & Bull, 2002*).

BF based deception detection techniques use cognitive demand induced BF suppression to indicate deception. Numerous studies demonstrate that increases in cognitive demand cause a reduction in BF (e.g., *Drew, 1951*; *Siegle, Ichikawa & Steinhauer, 2008*). In addition to BF suppression during lying, liars may also display a flurry of compensatory blinks after lying (*Leal & Vrij, 2008*). Importantly, truth tellers in that experiment exhibited an increase in BF during the relevant questioning period. While BF is correlated with many physiological and emotional states, this effect may be partially explained by the accusatory subject matter of the questions delivered during that period.

## Detecting deception about a third party

The majority of deception research has focused on subjective deception, or deception related to a personal transgression (*Iacono, 2000*). However, recent work has broadened the literature to include lying about characteristics of a third party (*Bhatt et al., 2009*; *Leal et al., 2011*; *Meijer et al., 2007*; *Shah et al., 2001*). Deception regarding a third party may differ characteristically from subjective deception because the stakes may be perceived as lower. Response times (*Haque & Conway, 2001*) indicate that recalling autobiographical information is more demanding than semantic recall, however the literature is unclear regarding differences in cognitive demand during deception about these types of information. Establishing these characteristics is important, since detecting deception about person recognition or familiarity may provide a means for establishing group affiliation, which in particular is not reliably detectable via traditional polygraph tests (*Sullivan, 2007*).

While comparative cognitive demands associated with discussing these topics truthfully and deceptively are not well understood, a number of studies explore deception about a third party using other means. In a novel experiment, *Leal et al. (2011)* asked participants to participate in mock espionage mission similar to the mock crime scenarios used by *Lykken (1957)*, *Lykken (1959)* to validate the GKT. Participants were briefed by one of the experimenters who revealed personal characteristics (i.e., hobbies), and later they were asked to identify and describe this experimenter from amongst a set of photographs. Cognitive demand as rated by observers, as well as differences in gaze direction, allowed discrimination between liars and truth tellers.

### Hypotheses

First, differences in BF between liars and truth tellers should be similar to those described in cases of subjective deception by *Leal & Vrij (2008)*. Second, liars are expected to exhibit a reduction in BF while lying, followed by a compensatory flurry of blinks. Finally, truth tellers are expected to exhibit an increase in BF while answering the mission relevant questions.

## METHOD

### Participants

The Saint Joseph's University (Philadelphia, PA) IRB board approved (IRB 2012-12) 34 participants (25 female, 9 male, age 18–21) for the experiment from the undergraduate population enrolled in introductory level psychology classes. All participants signed informed consent forms and were informed of the physiological measures, though they were blind with regard to the specific aspects of the electrooculography (EOG) data used in the analysis. One participant was excluded due to a failure to adequately follow the directions of the experiment and another participant exhibited an exceptionally low BF ($D_i = 2.16$). The sample used in the analyses therefore consisted of 23 females and nine males after exclusions.

### Data collection and analysis

Eyeblinks were monitored using an Apple iSight camera mounted on a modified Yukon Advanced Optics Inc. night vision head mount kit and positioned roughly 2 inches from the eye. For convenient analysis, blink frequencies were also recorded via EOG using AD Instruments' PowerLab 26T and the LabChart Pro v. 7 software package. Three electrodes, one on the orbicularis oculi muscle, one on the frontalis muscle, and a third on the ear as ground monitored eyeblinks in accordance with the protocol outlined by *Conduit (2012)* for monitoring blink amplitude. Since the present study is concerned only with quantifying blink *frequency*, no electrodes were placed to monitor more subtle eye movements. An Apple Macintosh iMAC computer was used for collecting and analyzing data. Blink data was recorded continuously, and quantification began immediately after the interviewer read the question and continued until the participant's response terminated. Blinks were quantified manually from concurrent recorded video of the eye from which EOG data was recorded. During this quantification, the experimenter was

blind to the participant's experimental condition. The EOG data were amplified (using the default sampling rate of 1 KHz) and filtered (Range = 2 mV, Low Pass = 10 Hz, High Pass = .5 Hz) and normalized in terms of *SD*. While slightly more liberal than other filters recommended for similar electrode placement (e.g., *Wissel & Palaniappan, 2011*, in which the authors recommend a filter with cutoff frequencies of 1 and 5 Hz), these parameters provided a smooth baseline with little noise and clearly discernible peaks. Peaks with an amplitude at least 4 *SD* higher than baseline activity indicated eyeblinks. There was no difference between blink occurrences recorded manually from video, or using EOG.

BF data was collected for each participant in four experimental periods: two baseline periods (at the beginning and end of the interview, during which the participants answered personal questions), a target period (containing the mission relevant questions), and a target offset period defined as the 6 s period following the target period (as observed by *Leal & Vrij, 2008*). By participant, for each experimental period, BF was quantified as the number of blinks in that period divided by the mean number of blinks exhibited in the two baseline periods. This value provides a measure of percent deviation in BF from baseline. Percent deviation scales the frequencies to account for individual differences in BF (*Leal & Vrij, 2008*). Results will be described in terms of this percent deviation metric.

## Procedure

The experimental protocol was an immersive mission based scenario similar to that used by *Leal et al. (2011)* and *Leal & Vrij (2008)*. The protocol consisted of a briefing and an interview. Participants arrived at a room in an academic building and received a briefing from one of the experimenters posing as a friendly agent, then went on a mock mission to deliver a package to a second room in the same building. There, participants would be interviewed by another experimenter role playing an anonymous agent. Participants were instructed to tell the truth to the interviewer or lie based on the interviewer's response to a challenge question. Correctly answering this challenge question would indicate to the participant that the interviewer is friendly, and that the participant should be entirely truthful. An incorrect response to the challenge question by the interviewer would indicate that the interviewer is an enemy agent to whom the participant should lie about all details of the mission. Participants were randomly assigned to the lying and truth telling conditions, leaving 15 truth tellers and 17 liars after exclusions.

During the briefing in the first room, participants were informed that the agent delivering the briefing was (1) a Saint Joseph's University graduate student, (2) did not receive his/her bachelor's degree from Saint Joseph's University, and (3) enjoys running. After the briefing, participants were sent to deliver the package to the second room. Participants were briefed by a male or female experimenter, and $t$ tests revealed no significant effect of briefer gender, all $t(30) < .31$, all $p > .75$, or participant gender, all $t(31) < 1.64$, all $p > .1$, on the experimental variables.

Upon arrival at the second room, participants were prepared for the EOG analysis and interviewed by another experimenter. Participants were interviewed through a one way

mirror and the voice of the interviewer was modified using a voice distortion microphone. This protocol was adopted to eliminate variance attributable to the interviewer's gender and body language. The interview consisted of two periods of free recall, the target period consisting of mission relevant questions, and a 6 s period immediately following the target period (target offset period). The interviewer waited 10 s between questioning periods.

During the baseline periods, participants were asked to freely recall information regarding irrelevant subject matter. For one baseline period, participants were told, "Please take one minute to tell me about your favorite actor or actress." Participants were asked to include in their responses shows or movies in which this person has acted, this person's on screen characters, and why they like this actor or actress. During the other baseline period, participants were asked to describe their favorite food, and to specifically address its national origin, whether there are any local restaurants in which to eat it, and whether they like it for its nutritional value or just for the taste. Baseline period content was counterbalanced to eliminate order effects. Despite one baseline period requiring description of a food and one of a person, there were no significant differences in BF between baseline periods, indicating that each baseline recall task was similarly demanding (data not shown). Likewise, both liars ($M = .59, SD = .22$) and truth tellers ($M = .50, SD = .13$) exhibited similar BF (per second) during the baseline periods, $t(30) = -1.46, p = .15$.

During the target period, participants answered mission relevant questions about the agent who delivered the briefing. The questions were:

1. Who sent you?

2. What does this person look like?

3. What does this person do for a living?

4. Did this person earn his or her bachelor's degree at SJU?

5. Does this person have a hobby?

For the three questioning periods (i.e., the two baseline periods and the target period), the experimenter delivered the questions, without breaks, then allowed the participants to freely recall the information and respond. In cases where participants' responses lasted less than 15 s, the experimenter prompted the participant to elaborate and continue. Participants were allowed to speak for up to 120 s. After questioning, participants completed a 7-point motivation Likert scale ("How motivated were you to do well in the interview?") and were debriefed.

## RESULTS

No between group differences were found in response length during the "favorite actor," "favorite food," or target period for liars and truth tellers, all $t(30) < .81$, all $p > .42$. Response length data for each group during the experimental periods are available in

**Table 1 Response lengths for liars and truth tellers during experimental periods.**

| Experimental Group | "Favorite Food" | "Target" | "Favorite Actor" |
|---|---|---|---|
| Liars | $M = 37.76, SD = 20.05$ | $M = 38.47, SD = 14.45$ | $M = 31.32, SD = 14.17$ |
| Truth Tellers | $M = 43.9, SD = 27.80$ | $M = 33.03, SD = 15.58$ | $M = 32.70, SD = 19.53$ |

Table 1. All participants reported a high degree of motivation according to the Likert scale ($M = 6.00, SD = .73$) and condition assignment did not affect participants' reported motivation to perform well in the interview, $t(29) < .001, p > .99$.

A 2 (Veracity: lying vs. truth telling) × 2 (Experimental Period: target period deviation vs. target offset deviation) factorial ANOVA, with repeated measures on the second factor, was conducted to assess differences in BF modulation patterns between liars and truth tellers. The analysis revealed a main effect for Experimental Period, $F(1, 30) = 16.17, p < .001, \eta p^2 = .35$, such that participants exhibited a higher BF during the target period ($M = .98, SD = .21$) than the target offset period ($M = .71, SD = .36$). The analysis also found an interaction effect of Veracity x Experimental Period, $F(1, 30) = 6.00, p = .020, \eta p^2 = .167$. The Veracity x Experimental Period interaction effect indicated that liars displayed a significantly suppressed BF during the target period ($M = .85, SD = .13$) compared to the increase exhibited by truth tellers ($M = 1.12, SD = .19; d = 1.66$), and less BF reduction during the target offset period ($M = .74, SD = .36$) compared to truth tellers ($M = .66, SD = .34; d = .23$; Fig. 1). Followup $t$-tests indicated significant differences in target period deviation for both liars, $t(16) = 4.56, p < .001$, and truth tellers, $t(14) = -2.27, p = .039$.

MANOVA identified between groups differences in the predictor variables, the target and target offset deviation scores, based on veracity as the independent variable. The data satisfied the assumption of homoscedasticity using Box's M. Hotelling's Trace revealed a significant multivariate effect of veracity on the dependent variables, $F(2, 29) = 9.91, p = .001, \eta p^2 = .406$. Univariate ANOVAs showed a significant effect of veracity on the target deviation score, $F(1, 30) = 19.95, p < .001, \eta p^2 = .40$, but an insignificant effect on target offset deviation, $F(1, 30) = .38, p = .54, \eta p^2 = .01$. Target period deviation from baseline was therefore retained as the sole predictor for discriminant analysis.

Discriminant analysis tested the capability of target period deviation to discriminate between liars and truth tellers. The discriminant function incorporating the predictor was significant, $\chi^2(2) = 15.04, p < .001$, with 81.3% of cases correctly classified. Using the discriminant function, 88.2% of liars and 73.3% of truth tellers were correctly classified, indicating that the function favored sensitivity over specificity. Because of the small sample size, the data were cross validated to check external validity using a jackknife procedure (*Lachenbruch, 1967*) which is appropriate for small sample sizes (*Stevens, 2009*). The original and cross validated classification results are shown in Table 2, and the significance and power of the discriminant function are provided in Table 3. Cross validation resulted in the misclassification of one truth telling participant.

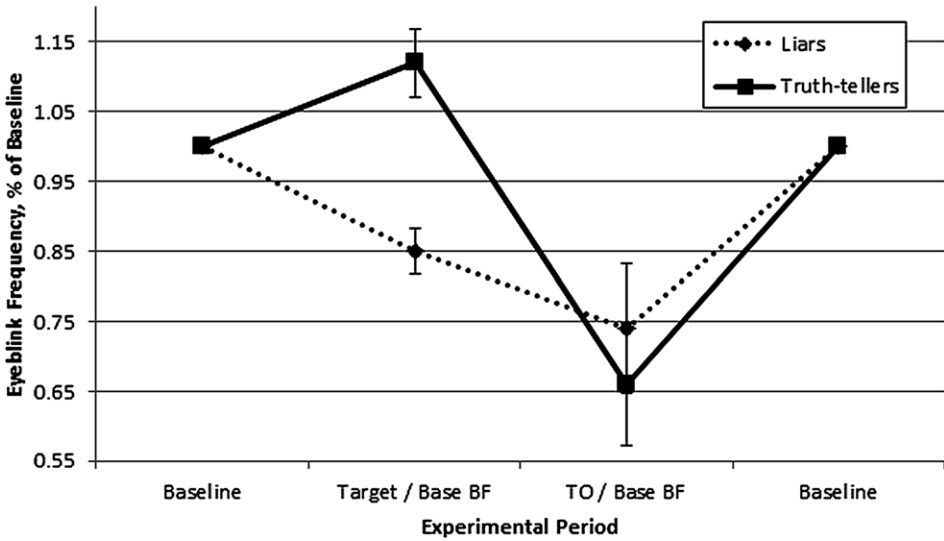

**Figure 1 Results of 2 (Veracity: Lying vs. Truth-telling) × 2 (Experimental Period: Target vs. Target Offset) ANOVA.** BF for each group across experimental periods quantified as percent change from baseline.

**Table 2 Classification table for percent-change scores between experimental periods.**

| Actual | Predicted veracity - Original | | |
|---|---|---|---|
| | Lying | Not lying | Total |
| Lying ($N$) | 15 | 2 | 17 |
| Not lying | 4 | 11 | 15 |
| Lying (%) | 88.2 | 11.8 | 100 |
| Not lying | 26.7 | 73.3 | 100 |
| **Actual** | **Predicted veracity – cross-validated** | | |
| | Lying | Not lying | Total |
| Lying ($N$) | 15 | 2 | 17 |
| Not lying | 5 | 10 | 15 |
| Lying (%) | 88.2 | 11.8 | 100 |
| Not lying | 33.3 | 66.7 | 100 |

**Notes.**
81.3% of original grouped cases correctly classified. 78.1% of cross-validated grouped cases correctly classified.

**Table 3 Significance of the discriminant function predicting veracity, and discriminating power of the discriminant function.**

| Wilk's Lambda | $X^2$ | d.f. | Significance |
|---|---|---|---|
| .601 | 15.04 | 1 | $<.001$ |
| Eigenvalue | Percentage of variance | Canonical correlation | |
| .665 | 100 | .632 | |

# DISCUSSION

One goal of the present study was to replicate the findings of *Leal & Vrij (2008)* in which, during free recall, liars exhibited suppressed BF during the target period followed by a compensatory flurry of eyeblinks, whereas truth tellers exhibited an increase in BF during the target period. The present study found a similar difference in BF between groups during the target period; however, neither group exhibited a compensatory flurry of eyeblinks during the target offset period (Fig. 1).

Though truth tellers' BF dynamics matched the findings of *Leal & Vrij (2008)*, there is a possible alternative to the anxiety explanation provided in that study that is congruent with the cognitive demand hypothesis. Since BF reflects state cognitive demand, it is possible that the increase from baseline observed in truth tellers during the target period is attributable to a state of reduced cognitive demand. Recalling the recently acquired semantic information truthfully was perhaps less demanding than retrieving autobiographical information to answer the baseline questions. This is supported by reaction time studies on autobiographical versus semantic recall (*Haque & Conway, 2001*). Therefore, it is possible that BF changes in truth tellers are the result of cognitive demand changes rather than anxiety or another emotional response.

The differences observed by *Leal & Vrij (2008)* may also be attributed to the content of the experimental periods used in that study; during the target period, participants were given no specific instruction, while their behaviors during the baseline periods were directed. In addition, participants in the lying and truth telling conditions engaged in different behaviors; liars committed a mock crime whereas truth tellers did not. If truth tellers' actions during the target period were less complex than their directed behaviors during the baseline periods, then recalling the target period information may have been less demanding.

The sharp BF reduction during the target offset period in the present study contradicts the findings of *Leal & Vrij (2008)*, who noted a flurry of compensatory blinks in liars. For this result, three proposed explanations follow. First, capturing that information may be exceptionally difficult due to the short window in which the flurry would occur. Frequency data for the target offset period is calculated over a 6 s period, which is significantly shorter than the other experimental periods. Second, these inconsistent results may be the result of differences in experimental protocol, such that a view of the interviewer, present in the study by *Leal & Vrij (2008)* but obscured in the present study, is necessary to induce these compensatory blinks. The final possibility is that the BF reduction observed in the present study is due to the fact that participants were not speaking or listening during this period. This possibility is supported by other literature on blink BF dynamics (e.g., *Karson et al., 1981*), which suggest that BF during silence is significantly lower than during speech or listening. These results, taken together, seem to indicate that the post-questioning compensatory flurry of blinks does not always follow deceptive responses, and depends heavily on other factors.

The second goal of the present study was to evaluate the BF modulation during a free recall test for application beyond subjective deception, to deception regarding a third

party. Validating the method in this way indicates greater robustness required for application, and an advantage over GKT-based techniques. Prior research using BF (*Leal & Vrij, 2008*) focused on recent subjective events, whereas the present study primarily focused on semantic details about a third party, though the target period did include a question regarding the purpose of the participant's "mission." BF characteristics were similar in the present study to those observed by *Leal & Vrij (2008)* for subjective deception.

While the results obtained herein indicate that it is possible to discriminate between liars and truth tellers, studies in this area do not speak to the fact that is perhaps more important in application: detecting deception within subjects. Since BF does not indicate deception, but rather cognitive demand, these techniques rely on between group comparisons to make causal inferences regarding the cause of the blink frequency suppression. Future research in this area should seek to explore BF dynamics associated with varying question content. Cognitive task analysis of common interrogation questions may aid in identifying analogue questions to serve as baselines (i.e., questions requiring similar cognitive demand to answer).

Additionally, certain ecological validity issues remain unresolved. The present study did not incorporate any meaningful interval between encoding of information (briefing) and testing (interview). *Carmel et al. (2003)* demonstrated that intervals as short as one week can significantly impair accuracy of other tests of deception (specifically the GSR based GKT) employed in experimental conditions. In addition, if liars are allowed to construct and rehearse an alibi, this would likely reduce cognitive demand as the fabrication component of deception would be removed. Therefore, the accuracy obtained in the present study should not be considered externally valid.

## CONCLUSIONS

Results of the present study suggest that a technique measuring BF reduction during deception, presumably resulting from increased cognitive demand, is sufficiently robust to detect deception when the suspect does not have a view of the interviewer, and when the suspect is asked about a third party. Because BF data can be collected surreptitiously using webcams and hidden cameras, and analyzed either in real time or post-hoc from recorded video, BF based techniques warrant consideration. However, there exist a number of hurdles to application that appear intrinsic to BF based techniques.

In the absence of between subject comparisons, it is perhaps not possible to definitively attribute BF suppression to deception. To the extent that it is possible to ameliorate this shortcoming, baseline content must be carefully developed and selected so as to be similarly demanding as truth telling in order to detect BF reductions indicative of lying. Manipulating the content of baseline and target questions could improve the sensitivity and specificity of the test. Furthermore, tests such as the GKT benefit from repetition of target questions (*Ben-Shakar & Elaad, 2002*), so perhaps multiple presentations of target questions, changed slightly as to require the fabrication of new responses, would also increase classification accuracy. While the technique offers several advantages to

traditional physiological methods for lie detection, additional research is required to determine if it is suitable for detecting deception within subjects.

## ACKNOWLEDGEMENTS

This work was completed to satisfy the requirements of a MS thesis. All materials and space used for this work are property of Saint Joseph's University, Philadelphia, PA. The author thanks Victoria Kurzeja for her participation as a confederate during this study. Additionally, the author thanks Ashley L. Adams, Donald S. Leitner, Philip Schatz, and Alex J. Skolnick for their input on this manuscript.

### Funding

This work was not funded through any external sources.

### Competing Interests

The author declares that he has no competing interests.

### Author Contributions

- Brandon S. Perelman conceived and designed the experiments, performed the experiments, analyzed the data, contributed reagents/materials/analysis tools, wrote the paper.

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
