# Peer review of "Detecting deception via eyeblink frequency modulation"

_PeerJ, doi:10.7717/peerj.260_

## Round 0.1 · original submission · Major Revisions

Dear Author,Please do the necesssary major revisions for the manuscript especially those raised by the first peer reviewer.

Reviewer 1 ·

Basic reporting

-The author uses the abbreviation “BF” for blink frequency. This abbreviation is introduced on line 64, but not defined until line 82. Please define the first time it is used.

-In general, the introduction was hard to follow. Improving transitions between sections would help the reader follow along. It may also be helpful to change the order of some of the information presented in the introduction (e.g. discussing cognitive demand hypothesis and why this applies to deception detection, then going into specifics about how blinks are measured and previous findings).

-The author makes the argument that an advantage of BF as a measure of deception is that it is possible to detect deception using “distance technology”. However, distance technology is not clearly defined (is it just web cams, or how else could this be used?), and the applications/benefits of this are not spelled out for the reader. Fleshing this out in the first paragraph of the introduction section would be helpful. The author already argues that current physiological measures aren’t great because they have to be attached to the subject and subjects are aware of the analysis. It will be important to discuss the alternative type of measurement here (covert cameras to detect BF, perhaps). This argument should also be carried through to the discussion to further emphasize the importance of this study in validating a covert technique to aid in deception detection.

-Throughout the Results section especially, I found the terms used to be cumbersome (e.g. line 198-199 “Target Period BF /Baseline Mean BF”). Couldn’t this be called “Target BF”. If the method for baseline correcting BF is clearly explained prior to this section, that could help streamline this language. This should also be consistent throughout the paper (e.g. in line 216, this same construct is referred to as “target over baseline percentage score”).

-Line 205 states, “an interaction effect of Veracity Experimental Period”, but should read “an interaction effect of Veracity X Experimental Period”

-Grammar and sentence structure could be better in lines 211-219.

Experimental design

-The author states that EMG was used to measure eyeblinks and then go on to describe that one electrode each was placed above and below the eye. This form of measurement is not EMG. In order to record EMG, two electrodes must be placed along the same muscle fibers to assess muscle contractions. What the author is describing appears to be EOG, a measurement of polarization differences across the eye, which will detect vertical eye movements rather than muscle contractions. This should be corrected in the manuscript.

-The author states (line 149), “Recordings of blinks began immediately after the interviewer read the question and continued until the participant’s response terminated”. However, they also say that BF was baseline corrected. It is unclear to me when baseline was measured and for how long. The authors later describe asking personal questions (e.g. describe favorite actor, line 173), but it is unclear to me whether this serves as the baseline or if baseline BF is measured at rest.

-Please provide specific formula for calculating baseline corrected BF. Is it a percent change score or something else?
-It is not clear to me whether EOG or camera data (or both) were used in the analyses. Were there any discrepancies between the two measurement methods and if so, how was this handled?

-Line 166-172, please provide additional information on the timing of this task, whether there were standard breaks between each question asked during the target period, and whether the length of each period was controlled in some way by the researchers.

Validity of the findings

-One significant confound of this study is that during the target period, participants are asked to answer questions about another person, while during the baseline they answered questions about their own interests. In order to rule out self-reference vs. other-reference as a potential confound, the author could examine differences between baseline and target periods for the truth-tellers, as these would not be expected to differ if BF is modulated solely by veracity. If they do differ (it’s hard to tell based on the figure), then it begs the question of whether BF in this case is truly a measure of cognitive demand related to deception, or if it’s capturing something else entirely (self-reference or task demands). The author does not present statistical tests to examine these questions (e.g. t-tests to compare target BF to baseline BF in each condition), although they do state in the discussion section that truth tellers exhibited an increase in BF during the target period (I assume as compared to baseline?) and discuss some reasons why cognitive demand may have been higher during baseline than the target period. These possible alternative explanations should be examined further.

-Lines 255-265, the author provides possible explanations for the lack of BF increase during target offset. Although speculation is acceptable, citations should be provided to support at least a portion of these claims when possible.

Additional comments

This is an interesting study examining blink frequency as a measure of deception detection with respect to a third party. Based on the current manuscript, I do not feel I have adequate information to fully evaluate the quality of study design and data analysis/interpretation. I also found areas of the paper hard to follow and believe improvements to wording and information flow could significantly strengthen this manuscript. If the author is able to address my concerns and provide appropriate information to enable further evaluation of the design, data scoring, and analysis techniques, I would be pleased to review again.

Reviewer 2 ·

Basic reporting

The study appears to be a replication of a previous study by Leal & Vrij (2008). This author found out that during lying, the subjects’ blink rate decreased, but did not observe increase in the blink rate as found in the earlier study.

Experimental design

The study is accurately founded, carried on and reported.

Validity of the findings

Validity of the findings has been adequately discussed. The author himself brings up a potential issue in the end of the discussion: maybe the acute need for fabrication of facts, rather than emotional load (=lying as such), in the ‘lying’ condition is what actually causes the observed difference in blinking frequency.

Additional comments

Minor issues:
1. The method used to record the eyeblinks with the mentioned electrode montage and filter parameters is usually not called EMG, which is considerably faster in frequency.
2. There is only one author but there are many references to ‘us’ in the text. For instance, “We expect differences in BF between….” on line 118.
3. The abbreviation BF is introduced in the beginning of the introduction but opened up not until line 82.

---

## Round 0.2 · Minor Revisions

The previous peer reviewer still requires a bit of explanation to be highlighted:Read statement I raised the question about the use of the term EMG (and so did reviewer 1). I only meant that if the mentioned filter parameters have been used, we are not talking about muscle activity recordings (EMG), since motor unit activity is too fast to be detected in the frequency range of 0.5 to 10 Hz, which appears to be the bandpass the author used, (by the way, sampling rate should also be mentioned). As such, there is no doubt in my mind that the author would not have been reliably recorded eyeblinks. Not sure what to call the method, though.I think one should elaborate on the methodology.

Reviewer 1 ·

Basic reporting

No comments

Experimental design

No comments

Validity of the findings

No comments

Additional comments

In the previous review, I raised the question about the use of the term EMG (and so did reviewer 1). I only meant that if the mentioned filter parameters have been used, we are not talking about muscle activity recordings (EMG), since motor unit activity is too fast to be detected in the frequency range of 0.5 to 10 Hz, which appears to be the bandpass the author used, (by the way, sampling rate should also be mentioned). As such, there is no doubt in my mind that the author would not have been reliably recorded eyeblinks. Not sure what to call the method, though.

Otherwise, I have no further comments.

---

## Round 0.3 · accepted · Accept

Thank you for the revisions made to the manuscript leading to it's acceptance